# 3H-1,2-Dithiole-3-Thione Protects Lens Epithelial Cells against Fructose-Induced Epithelial-Mesenchymal Transition via Activation of AMPK to Eliminate AKR1B1-Induced Oxidative Stress in Diabetes Mellitus

**DOI:** 10.3390/antiox10071086

**Published:** 2021-07-06

**Authors:** Tsung-Tien Wu, Ying-Ying Chen, Chiu-Yi Ho, Tung-Chen Yeh, Gwo-Ching Sun, Ching-Jiunn Tseng, Pei-Wen Cheng

**Affiliations:** 1Department of Ophthalmology, Kaohsiung Veterans General Hospital, Kaohsiung 81362, Taiwan; ttwu@vghks.gov.tw (T.-T.W.); yychen@vghks.gov.tw (Y.-Y.C.); 2School of Medicine, National Yang-Ming University, Taipei 11221, Taiwan; 3Department of Optometry, Shu-Zen Junior College of Medicine and Management, Kaohsiung 81362, Taiwan; 4Department of Medical Education and Research, Kaohsiung Veterans General Hospital, Kaohsiung 81362, Taiwan; hochiuyi1987@gmail.com (C.-Y.H.); jctseng@vghks.gov.tw (C.-J.T.); 5Department of Biomedical Science, National Sun Yat-Sen University, Kaohsiung 80424, Taiwan; 6Department of Internal Medicine, Division of Cardiology, Kaohsiung Veterans General Hospital, Kaohsiung 81362, Taiwan; tcyeh9@gmail.com; 7Department of Anesthesiology, Kaohsiung Medical University Hospital, Kaohsiung Medical University, Kaohsiung 80708, Taiwan; gcsun39@yahoo.com.tw

**Keywords:** AMPK, 3H-1,2-dithiole-3-thione (D3T), AKR1B1, diabetic cataracts, epithelial-mesenchymal transition (EMT)

## Abstract

Studies demonstrated that the receptor of advanced glycation end products (RAGE) induced epithelial-mesenchymal transition (EMT) formation in the lens epithelial cells (LECs) of diabetic cataracts. This work investigated how 3H-1,2-dithiole-3-thione (D3T) reduces EMT formation in LECs of the fructose-induced diabetes mellitus (DM). LECs were isolated during cataract surgery from patients without DM or with DM. In a rat model, fructose (10% fructose, eight weeks) with or without D3T (10 mg/kg/day) treatment induced DM, as verified by blood pressure and serum parameter measurements. We observed that the formation of advanced glycation end products (AGEs) was significantly higher in epithelial human lens of DM (+) compared to DM (−) cataracts. Aldose reductase (AKR1B1), AcSOD2, and 3-NT were significantly enhanced in the rat lens epithelial sections of fructose-induced DM, however, the phosphorylation level of AMPK^T172^ showed a reversed result. Interestingly, administration of D3T reverses the fructose-induced effects in LECs. These results indicated that AMPK^T172^ may be required for reduced superoxide generation and the pathogenesis of diabetic cataract. Administration of D3T reverses the fructose-induced EMT formation the LECs of fructose-induced DM. These novel findings suggest that the D3T may be a candidate for the pharmacological prevention of cataracts in patients with DM.

## 1. Introduction

Diabetes mellitus (DM) can lead to pathologies in many tissues in the eye structure, bearing the nature of systemic chronic metabolic disease and microangiopathic characteristic [1]. Diabetes has been shown to be a risk factor for age-related (AR) cataracts; it has been predicted that by the year 2030, >400 million people in the world will have diabetes [2]. An aging population and longer life expectancy mean that DM will exceed 33% of the 65 years old and elder population by 2050, and a global increase in AR cataracts is also expected [3]. In general, DM patients show tendencies of up to five times more likely to develop cataracts, particularly during early stage [4,5]. At present, other than removal of the cataract, surgeries to treat cataracts continue to increase every year [6]. Although technological and medical advancement have greatly improved surgical outcome, the quality of vision in DM patients who undergo cataract removal remains unpredictable. Many studies showed complications from cataract surgery in DM patients, such as diabetic macular oedema (ME), postoperative ME, diabetic retinopathy progression, and posterior capsular opacification [2].

Evidence indicated that clinical antidiabetic drug treatment did not yield any clear association with lower cataract risks [7]. Furthermore, studies found a statistically significant positive trend for association between diabetes duration and risk of cataract, and yielded increased relative risks for development of a cataract with higher HbA1c level [7]. There is accumulating evidence that dietary sugar-mediated generation of advanced glycation end products (AGEs) and activation of the receptor for advanced glycation end products (RAGE), both of which are senescent protein derivatives that result from the auto-oxidation of glucose and fructose, contribute to metabolic syndrome [8,9]. Moreover, impaired antioxidant ability of epithelial lenses caused increased susceptibility to oxidative stress in diabetic patients. Predominantly, the antioxidant enzyme, superoxide dismutase (SOD) in the lens degrades superoxide radicals (O_2_^−^) into H_2_O_2_ and oxygen. As a result, decreased activity of antioxidant enzymes, uncontrolled activation of AR, and accumulation of AGEs have been considered as signs hallmarks for cataracts in diabetic patients [10,11]. 

Previous studies showed that the NADPH-dependent aldo-keto reductase, also known as AKR1B1 in human, is a catalyst for converting glucose to sorbitol in the polyol pathway and was suggested to be critical for cataract development [12]. Mechanistically, AKR1B1 is involved in retinopathy, cataractogenesis, neuropathy, and cardiovascular disease, which are part of diabetic complications [13]. The inhibition of AKR1B1 has been an attractive approach for the treatment and management of diabetic complications. Evidence showed that AKR1B1 overexpression in the lens of transgenic mice led to anterior subcapsular cataracts development, even in the absence of diabetes and hyperglycemia [14]. Additionally, it was demonstrated that AKR1B1 inhibitors were capable of suppressing lens epithelial cells’ proliferation [14]. Apart from AKR1B1 involvement, epithelial to mesenchymal cells transition (EMT) in lens’ epithelial cells is also suggested to cause posterior capsule opacification (PCO), a common post-surgical complication resulted from cataract surgery [15]. 

The intermediary metabolism regulates glucose and fatty acid uptake in cells, and it requires the serine/threonine protein kinase, AMP-activated protein kinase (AMPK). Under various stress conditions such as glucose starvation, hypoxia, ischemia, and oxidative damage, AMPK is activated to alter cellular energy level [16]. In response to the aforementioned stresses, AMP:ATP ratio is changed through phosphorylation of Thr172 in the catalytic α subunit activation loop in the AMPK [17]. Since AKR1B1 was shown to promote diabetic complications and play important role in oxidative stress during diabetes, AKR1B1 inhibition was able to attenuate alcoholic liver disease by AMPK activation and oxidative stress modulation [18]. Chang et al. reported that AR knockout increased the expressions levels of AMPK, SIRT1, and ATP in both normal and fatty livers after major hepatectomy and IR injury [19].

3H-1,2-Dithiole-3-thione (D3T) is a member of the 1,2-dithiole-3-thione-compounds naturally occurring in cruciferous vegetables. Among all 1,2-dithiole-3-thiones, D3T is the most potent in inducing tissue defenses against oxidative and inflammatory stress [20]. D3T potentiates AGE-induced ROS formation suppressed by the inhibition of protein kinase C and NADPH oxidase [21]. We previously reported that AKR1B1 overexpression abolishes AMPK activation, thereby increasing acetyl superoxide dismutase 2 (AcSOD2) and RAGE-induced EMT, and is an important factor involved in the cataract pathogenesis [5]. 

In this study, we aimed to investigate whether D3T reduces EMT in the fructose-induced DM through an AKR1B1-enhanced ROS generation. We showed that fructose-induced DM, AKR1B1 overexpression would abolish AMPK activation, thereby increasing AcSOD2 and RAGE-induced EMT in the LECs, even without DCs. D3T downregulated AKR1B1-induced AMPK and caused an AcSOD2 imbalance, thereby decreasing EMT in the LECs, which is known to be a potentially important factor in the pathogenesis of cataracts in rats with fructose-induced DM. These novel findings suggest that D3T may be a candidate for the pharmacological prevention of cataracts in patients with DM.

## 2. Materials and Methods

### 2.1. Ethics Statement

The protocol for this study was reviewed and approved by the Institutional Review Board of the Kaohsiung Veterans General Hospital (Kaohsiung, Taiwan; IRB number: VGHKS17-CT5-10; VGHKS18-CT7-22).

This prospective study comprised patients who underwent phacoemulsification and intraocular lens implantation between February 2019 and February 2020 at Kaohsiung Veterans General Hospital. The study protocol followed the guidelines of the Declaration of Helsinki and was approved by the hospital’s institutional review board. After receiving a full explanation of the surgical procedures and possible complications, all patients provided written informed consent. All the data and specimens were collected and anonymized before analysis. The patients were selected based on the clinically confirmed nuclear cataracts of grade 2 or 3 according to the Lens Opacities Classification System III [17]. The patients were classified into 2 groups: patients without DM (Group 1) and patients with DM but without DR (Group 2).

### 2.2. Reagents

Experimental drugs including fructose and dimethyl sulfoxide (DMSO) were all obtained from Sigma-Aldrich (Sigma Chemical Co., St. Louis, MO, USA). D3T (Abcam, ab141925) was obtained from Abcam (Abcam Chemical Co., Cambridge, MA, USA). Primary antibodies, nitrotyrosine, and RAGE (sc-32757 and sc-8230, respectively, Santa Cruz, San Jose, CA, USA), N-cadherin and E-cadherin (610920 and 610182, respectively, BD Biosciences, San Jose, CA, USA), AKR1B1, MMP9, and Vimentin (GTX113381, GTX100458 and GTX100619, respectively, Genetex, Irvine, CA, USA), Phosph-AMPK α1,2^T172^ (44-1150G, Thermo Fisher Scientific, Eugene, OR, USA), and SOD2 (acetyl K68) (ab137037, Abcam, Eugene, OR, USA), and secondary antibodies, Alexa Fluor 568 and 488 goat anti-rabbit or mouse, and Alexa Fluor 488 donkey anti-goat, were used to stain the LECs. In addition, the VECTASHIELD^®^ HardSet™ antifade mounting medium with DAPI (H-1500, Vector, Burlingame, CA, USA) was used to mount the stained epithelial cell sections and counterstain them.

### 2.3. Animals

Sixteen-week-old male WKY rats were obtained from the National Science Council Animal Facility (NSCAF) (Taipei, Taiwan), and all animals were housed in facility approved by the Association for Assessment and Accreditation of Laboratory Animal Care, International (AAALACi). The animals were housed in pathogen free facility at temperature 23–24 °C, condition of 12 h light and 12 h dark cycle, provided with normal rat chow (Purina; St. Louis, MO, USA) and tap water ad libitum. SPF facilities maintained the animals in an environment free from certain infectious organisms that are pathogenic and/or capable of interfering with research objectives. All animal experiments were performed according to the Animal Research: Reporting of In Vivo Experiments (ARRIVE) guidelines [20,21] and procedures were approved by the Animal Research Committee and the Institutional Review Board of Kaohsiung Veterans General Hospital. To rule out experimental biases, drug treatments for the animals were conducted using the blind method, and all animals were included in this study.

The animals were acclimated to the housing environment for one week and then habituated to the indirect blood pressure measurement procedure for another week. For oral administration, animals were randomly assigned to three groups, each group contained six animals: (1) control group: WKY rats fed drinking water; (2) fructose group: consisting of WKY rats fed 10% fructose in drinking water for 8 weeks; (3) The fructose+D3T group: composed of WKY rats fed 10% fructose in drinking water and administered D3T (10 mg/kg/day) via gavage for 8 weeks. All animals in the experimental groups developed type 2 DM with no incidence of heart failure or sudden death.

### 2.4. Tissue Sample Collection

Based on the 2013 American Veterinary Medical Association (AVMA) guidelines, all animals were euthanized using 100% CO_2_. Death occurred within 2–5 min, followed by immediate removal of the lens, which was quickly frozen on dry ice. Tissues collected from identical experimental groups were pooled and stored at −80 °C.

### 2.5. Immunoblotting Analysis

To extract total protein from lenses, lenses were homogenized in lysis buffer composed of protease and phosphatase inhibitor cocktails for 1 h at 4 °C. The extracted protein concentration was quantified by BCA protein assay (Pierce), subjected to 7.5–10% SDS-Tris glycine gel electrophoresis, transferred to polyvinylidene difluoride membrane (GE Healthcare, Buckinghamshire, UK). The membrane was incubated in 5% non-fat skim milk in with Tween-20 buffer (10 mmol/L Tris, 150 mmol/L NaCl, and 0.1% Tween-20, pH 7.4) for 1 h. Primary antibodies: AKR1B1, MMP9, vimentin (GTX113381, GTX100458 and GTX100619, respectively, Genetex, Irvine, CA, USA), N- and E-cadherin (610920 and 610182, respectively, BD Biosciences, Franklin Lakes, NJ, USA), Phosph-AMPK α1,2^T172^ (44-1150G, Thermo Fisher Scientific, Eugene, OR, USA), and SOD2 (acetyl K68) (ab137037, Abcam, Eugene, OR, USA) were diluted in PBST at 1:2000 dilutions and incubated at 4 °C overnight. Peroxidase-conjugated anti-mouse or anti-rabbit secondary antibodies (1:10,000) were used. The result was visualized using ECL-Plus detection kit (GE Healthcare) and exposed to film. The developed films were scanned (photo scanner 4490, Epson, Long Beach, CA, USA), analyzed using NIH image densitometry analysis software (National Institutes of Health, Bethesda, MD, USA).

### 2.6. Blood Pressure Measurements

At week 0, before fructose or D3T treatment, systolic blood pressure of animals was measured using tail-cuff monitor (Noninvasive Blood Pressure System, SINGA, Taipei, Taiwan). Briefly, animals were stationed in the fixer for 30 min, with ambient temperature maintained at 34 °C. Six readings were obtained consecutively for each animal, wherein the maximal and minimal readings were discarded, and the remaining six readings were averaged. To ensure consistency, all systolic blood pressures of the animals were measured at the same time every day. 

### 2.7. Immunofluorescence Staining Analysis 

The samples were permeabilized with 0.5% Triton X-100 for 15 min, then blocked with 10% normal horse serum for 30 min. Afterward, the specimens were incubated with different primary antibodies in 10% normal horse serum at 4℃ overnight. Specimens were then treated with secondary antibodies in 10% normal horse serum for 2 h at 24–26 °C. Finally, the sections were mounted and counterstained with mounting medium which contains DAPI. The sections were analyzed using fluorescence microscopy and Zeiss LSM Image software (Carl Zeiss MicroImaging, DE Thüringen, Jena, Germany).

### 2.8. Statistical Analysis

The student’s *t*-test, Mann–Whitney U test, ANOVA, and Kruskal–Wallis one-way ANOVA were used to evaluate the correlation between DM and non-DM cataract patients. All statistical analyses were carried out on raw data using SPSS, version 20.0 (SPSS Inc, Chicago, IL, USA). Results were considered statistically significant at two-tailed value of *p* < 0.05. The data were represented as mean ± SEM. 

## 3. Results

### 3.1. Activated AGEs and Enhanced MMP9 Production Are Involved in DM (+) Cataract LECs Pathogenesis 

The accumulation of oxidized lens components and decreased efficiency of repair mechanisms can contribute to lens’ opacities or cataracts in chronic hyperglycemia and during aging [22,23]. Refraction and axial length were characterized in six DM (−) patients with cataracts as the control group (two males, four females), and DM (+) patients with cataracts (two males, four females) as experimental groups. The average age was 66.0 ± 3.5 years for the cataract patients and 66.8 ± 6.1 years for the DM patients (*p* = 0.87). Demographics and baseline clinical characteristics of the study participants are shown in Table 1. During EMT, members of the zinc-finger family of transcription factors, such as Snail, Slug, and Twist, and matrix metalloproteinases (MMPs) are upregulated, and this upregulation plays an important role in the development of fibrotic cataract phenotypes [24]. In addition, we investigated the connection of AGEs level in the LECs and EMT in the DM patients having cataracts. Immunofluorescence assay showed that AGE levels were significantly higher in DM patients’ LECs (Figure 1). Interestingly, MMP9 expression level was high in the LECs both DM(+) and DM(−) patients.

### 3.2. D3T Improves Fructose-Induced Hypertension, Does Not Prevent Fructose-Mediated Metabolic Defects

Blood pressure, fasting glucose, triglyceride, and high-density lipoprotein levels were measured (Table 2). Consistent with a recent report [25,26], our result showed significant increase in serum triglycerides in the fructose group compared to the control group. The fasting blood glucose level was increased, and direct high-density lipoprotein (dHDL) was decreased in the fructose group. The fasting glucose and triglyceride levels remained significantly higher in the fructose-D3T compared to control (159.7 ± 6.8 versus 98.7 ± 6.1 and 148.7 ± 9.0 versus 77.8 ± 7.8, respectively). In addition, the level of HDL was relatively higher in the fructose-D3T compared to the fructose group (74.8 ± 0.9 versus 58.8 ± 2.2). D3T addition prevented fructose-induced hypertension but showed no improvement for fructose-mediated metabolic defects.

### 3.3. D3T Reduces AKR1B1-Induced ROS Generation and Decreases EMT in the LECs of Rats with Fructose-Induced Type 2 DM

Several studies have demonstrated that overproduction of AKR1B1 in the LECs would undergo EMT that developed anterior subcapsular cataracts in vivo, even without diabetes and hyperglycemia [14]. Analysis of the DM (+) cataract group showed that MMP9 and AGE expression levels were significantly higher compared to the control group (Figure 1). As mentioned above, EMT is known to be one of the factors involved in the pathogenesis of cataracts. Therefore, we further examined the correlation between EMT and AKR1B1 in the LECs of rats with fructose-induced type 2 DM by immunofluorescence assay. As shown in Figure 2, fructose presence showed downregulation of AKR1B1, E-cadherin, and epithelial markers and upregulation of N-cadherin and mesenchymal markers; however, addition of D3T reversed these effects. These results indicated that D3T reduced AKR1B1-induced EMT expression in the LECs of rats with fructose-induced type 2 DM.

### 3.4. D3T Enhancement of AMP-Activated Protein Kinase (AMPK) Reduces ROS Production and SOD2 Acetylation in the Lens of Rats with Fructose-Induced Type 2 DM

Misra P and Chakrabarti R showed that enhanced AMPK activity improved type 2 diabetes metabolic disorder [27], and our previous studies demonstrated that AKR1B1 overexpression decreased AMPK activation, and increased AcSOD2 and RAGE-induced EMT in the LECs of DM patients with cataracts [5]. In addition, it was confirmed that oxidative stress plays an important role in the pathogenesis of metabolic syndrome [28]. The above results indicate that AMPK activation and superoxide accumulation are major signaling pathways in diabetics. D3T is reported to trigger tissue defense against oxidative and inflammatory stress [29]. Therefore, we investigated the effects of D3T, whether D3T can maintain the balance between the expression of p-AMPK^T172^ and another superoxide in the LECs of rats with fructose-induced type 2 DM. As shown in Figure 3, fructose groups exhibited downregulation of p-AMPK^T172^ and upregulation of 3-NT and AcSOD2; however, D3T addition reversed these effects (Figure 3A,B). In comparison to the fructose group, D3T increased the level of p-AMPK^T172^ and reduced 3-NT, a product of oxidative stress. We further examined the effect of fructose on the suppression efficiency of ROS products. Fructose increased AcSOD2 expression level that caused a loss of antioxidant SOD2 activity which normally helps remove superoxide radicals; however, D3T reversed these effects. 

### 3.5. D3T Reduces Snail/Slug Signaling Pathways to Ameliorate AKR1B1-Mediated Epithelial-to-Mesenchymal Transition through AMPK Activation 

EMT transcription factors, the well-characterized Snail-family proteins Snail (*SNAI1*) and Slug (*SNAI2*) play important roles in both physiological and pathological EMT [30]. D3T is known to reduce AKR1B1-induced EMT in the LECs of rats with fructose-induced type 2 DM. We investigated whether D3T was able to relieve AKR1B1-mediated EMT through the Snail/Slug signaling pathway. As shown in Figure 4, immunoblotting analysis of proteins extracted from the LECs demonstrated that D3T treatment inhibited the expression of Aldose reductase and AcSOD2 and increased AMPK levels in rats fed with fructose for eight weeks (Figure 4A). In fructose-fed animals, level of EMT-related transcription factors such as Snail and Slug were increased, whereas D3T administration reversed this effect (Figure 4B). However, D3T treatment decreased E-cadherin expression in rats fed with fructose for eight weeks (Figure 4C). These results indicated that D3T reduced AKR1B1-induced EMT via activating AMPK in the LECs derived from rats with fructose-induced type 2 DM.

## 4. Discussion

Diabetes is a major public health problem associated with an increased risk for cardiovascular disease, which is the leading cause of death worldwide [31]. Studies in yeast suggested that uncontrolled blood glucose elevation is the major risk factor for DM cataract development [32]. Lens’ opacity and oxidation [33], ROS formation, and sorbitol accumulation through AR conversion of glucose [34] have been linked to blood glucose elevation and lens protein glycosylation during cataract development. Since AGEs are found to play major role in lens degeneration, low level of AGE foods might be beneficial for counteracting cataract formation [35]. This study confirms that AGEs and MMP9 levels are significantly upregulated in DM (+) cataract patients (Figure 1). We previously showed that RAGE induced NADPH oxidase expression in the fructose-induced type 2 DM lens, even though there was no cataract formation [36]. Current evidence supports the view that cataract formation may involve EMT [37]. In regard to EMT, Du et al. reported that oxidative stress increases AR activation and AGE formation, driving EMT in early stages of diabetic cataract formation [37]. Furthermore, Korol at al. reported that MMPs are required for EMT in fibrosis of anterior subcapsular cataract (ASC) mouse models [24]. Moreover, AKR1B1 is an NADPH-dependent aldo-keto reductase, well-known as a catalyst for glucose conversion to sorbitol in the polyol pathway, which has been suggested to be essential for cataract development [12]. Furthermore, it was demonstrated that AKR1B1 inhibition led to downregulation of lens epithelial cells’ proliferation and EMT [14]. Interestingly, this study finds that AKR1B1 was capable of inducing EMT protein expression in the lens of rats with fructose-induced type 2 DM without developing cataract, and AKR1B1 treatment reversed the potentiation effect of D3T (Figure 2 and Figure 4). These results suggest that NADPH blockage by D3T prevented AKR1B1-induced oxidants’ generation, downregulated E-cadherin, and Snail/Slug signaling pathway. 

D3T, a cyclic sulfur-containing dithiolethione derived from cruciferous vegetables, protects cells from oxidative lesions by production of antioxidants, and this antioxidative effect is mainly through activation of the nuclear factor (erythroid-derived 2)-like 2 (Nrf2) signaling pathway [38]. As a transcription factor, Nrf2 regulates the expression of several target genes, including those encoding glutathione synthesizing enzymes, quinone reductases, glutathione reductase (GR), as well as NADPH-generating enzymes such as glucose-6-phosphate dehydrogenase (G6PD) [39]. AGEs utilize RAGE to modulate gene expression and mediate superoxide generation upon NADPH oxidase activation [40]. When Nrf2-mediated NADPH is downregulated, NAD(P)H oxidases (a major source of AGE-induced oxidative stress) seize this opportunity to increase ROS such as superoxide. In order to prevent cell damage by ROS, Nrf2 activator, D3T, induces GSH synthesis [41]. Global gene co-expression analysis in fasted animals (energy deficient animals) enabled the identification of AMPK as a potential Nrf2 kinase. AMPK plays a key role in sensing and modulating energy homeostasis by monitoring energy status (the AMP/ATP ratio) in the eukaryotic cells [42]. Wang S et al., studied the deletion of AMPKα2 enhanced NADPH oxidase subunit expression and increased oxidative stress in vascular endothelial cells [43]. Furthermore, Misra and Chakrabarti found that the severity of type II diabetes was effectively reduced via enhancing AMPK activity [27]. Moreover, Tsubota et al. showed that downregulated AMPK signaling pathway was involved in lens inflammation in the diabetic mice [44], indicating that AMPK activity is involved in accumulation of superoxide and other ROS. In addition, the inhibition of AKR1B1 attenuated alcoholic liver disease by activating AMPK and modulating oxidative stress [18]. Our results demonstrated that 3-nitrotyrosine (3-NT) and AcSOD2 levels were elevated significantly in LECs, whereas p-AMPKT172 level was reduced significantly in the lenses obtained from rats with fructose-induced type 2 DM. Additionally, D3T treatment enhanced AMPK activity, reduced ROS productions, and decreased acetylation of SOD2 in the lenses obtained from the rats with fructose-induced type 2 DM (Figure 3). Joo et al. also demonstrated that AMPKThr172 directly phosphorylates Nrf2S550 to facilitate its accumulation in the nucleus [45]. Their findings identified AMPK as a novel kinase which activates Nrf2 through nuclear accumulation, leading to the conclusion that AMPK is an energy-sensing enzyme in cells with increased defense mechanism against antioxidants for cell survival [46]. Collectively, these results suggest that strategies aimed at combating oxidative stress by downregulating the endogenous antioxidant defense system via AMPK may prevent cataract in patients with DM. Available scientific evidence suggests D3T as potential therapeutic agent for cataract associated with type 2 DM, since the D3T treatment can be administered through the eye drops. 

According to the International Diabetes Federation, 415 million people had diabetes in 2015 and this number is expected to reach 642 million by 2040, with >55% of cases expected to occur in Asia [45]. High incidence of diabetes poses serious public health and financial burdens. The annual medical cost for diabetes treatment is estimated to increase from USD 802 billion to USD 1452 billion in 2040 [47]. Therefore, identifying new and easy-to-access preventive compound(s) that can alleviate health complications associated with diabetes is imperative. Evidence indicated that clinical antidiabetic drug treatment did not yield any clear associations with lower cataract risks. In addition, our studies found that D3T reduces EMT in the fructose-induced DM through an AKR1B1-enhanced ROS generation. We also found that in fructose-induced DM, AKR1B1 overexpression abolished AMPK activation, leading to increased AcSOD2 and RAGE-induced EMT in the LECs in the absence of DCs. On the contrary, D3T downregulated AKR1B1-induced AMPK, causing an imbalance of AcSOD2, and reduced EMT in the LECs of fructose-induced DM rats. These novel findings suggest that D3T is a potential candidate for pharmacological prevention of cataracts in patients with DM.

## 5. Conclusions

In conclusion, AKR1B1 overexpression inhibited AMPK activation, thereby increasing AcSOD2 and RAGE-induced EMT in the LECs of diabetic cataracts. Regulation of AMPK activation is crucial for pathogenesis of diabetic cataracts (Figure 5B). AKR1B1 overexpression abolished AMPK activation, leading to increased AcSOD2 and RAGE-induced EMT in the LECs of rats with fructose-induced DM, even without diabetic cataracts. D3T downregulated AKR1B1-induced AMPK activation and AcSOD2 generation, leading to decreased EMT in the LECs (Figure 5D). These novel findings deepen our understanding of D3T as an important candidate for pharmacological intervention to treat cataracts in DM patients.

## Figures and Tables

**Figure 1 antioxidants-10-01086-f001:**
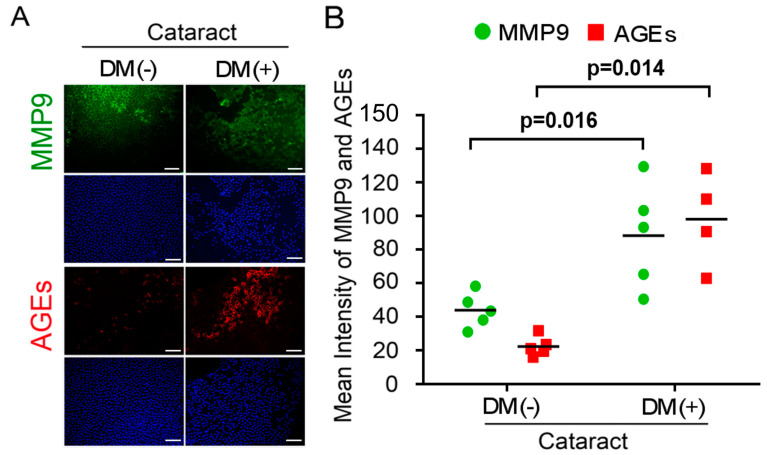
Expression of matrix metalloproteinase 9 (MMP9) and advanced glycation end products (AGEs) in the lens of DM (+) and DM (−) cataract patients. (**A**,**B**) Representative images of MMP9−positive (green), AGEs-positive (red) cells, and cell nuclei counterstained with DAPI (blue) in the lens of epithelial tissue from cataract DM (+) and DM (−) patients. Data represented mean ± SEMs (*n* = 6 per group, separate experimental groups in each figure).

**Figure 2 antioxidants-10-01086-f002:**
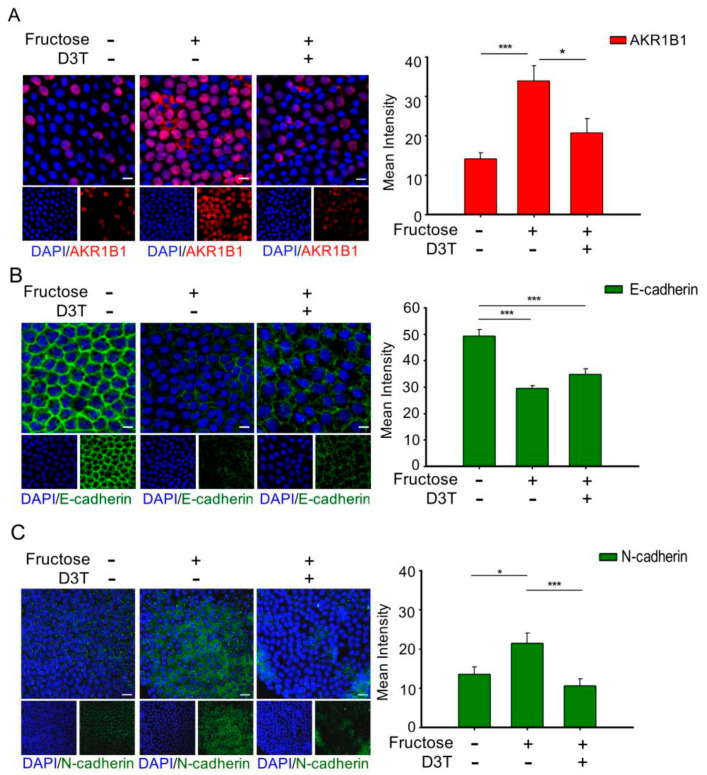
D3T reduces expression of AKR1B1 in the lens of rats with fructose−induced type 2 diabetes mellitus (DM). (**A**) Representative images showing AKR1B1−expressing cells (red) in the lens with or without systemic administration of fructose or D3T. (**B**,**C**) Representative images showing the expression of E-cadherin (green) in the cells of lens with or without systemic administration of fructose or D3T. Quantitative analyses for AKR1B1 and E-cadherin levels in the lenses of rats with fructose-induced type 2 DM after D3T administration were performed. All data represented as mean ± SEM (*n* = 6 per group, independent experiments in each figure). * *p* < 0.05; *** *p* < 0.001.

**Figure 3 antioxidants-10-01086-f003:**
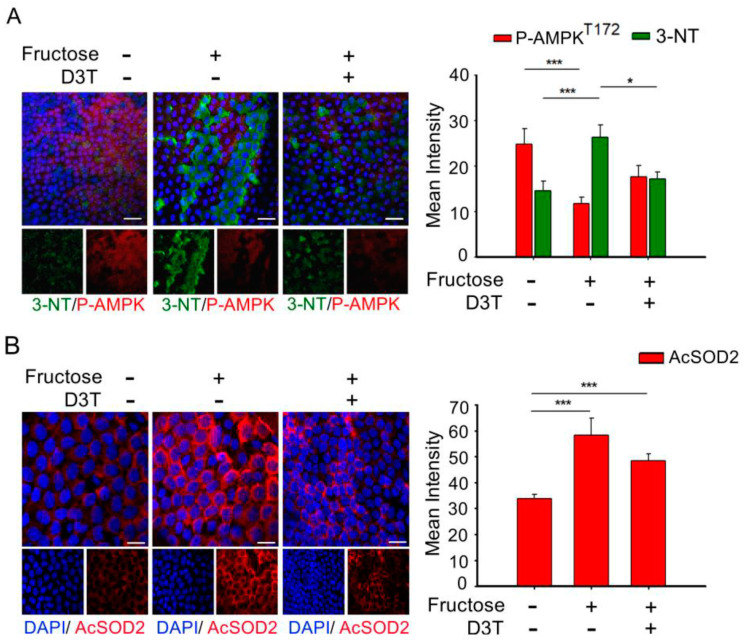
D3T reverses the effects of fructose−upregulated 3−NT and acetyl SOD2 in the lens of rats with fructose-induced type 2 DM. (**A**) Representative images showing the expression of p-AMPK^T172^ (red fluorescence) and 3-nitrotyrosine (3-NT, green fluorescence) in the lens with or without systemic administration of fructose or D3T. (**B**) Immunofluorescence assay was also performed for acetyl SOD2 (AcSOD2) (red) in the lens with or without systemic administration of fructose or D3T. All data represented mean ± SEM (*n* = 10 per group, independent experiments in each figure). * *p* < 0.05 and *** *p* < 0.001.

**Figure 4 antioxidants-10-01086-f004:**
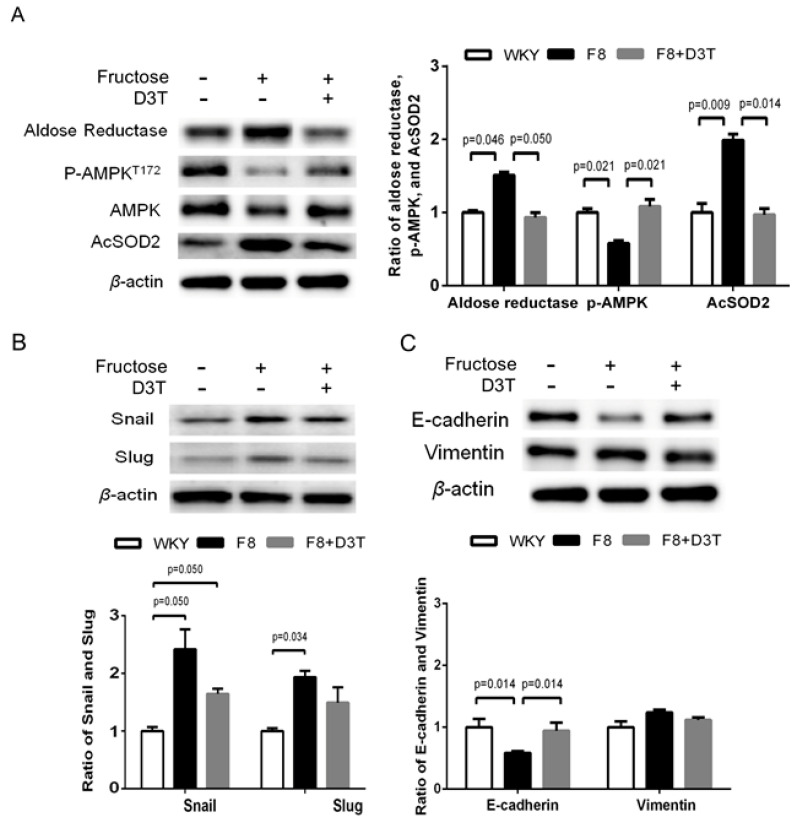
D3T reduces fructose−induced EMT progression in the lens of rats with fructose−induced type 2 DM. (**A**) Immunoblotting analysis depicting Aldose reductase, p-AMPK^T172^ and Ac-SOD2 expression in the fructose-induced type 2 DM lens with or without D3T administration. (**B**) Immunoblotting analysis showing Snail and Slug protein expressions in the frucTable 2. DM lens with or without D3T administration. (**C**) Protein expressions of E-cadherin and vimentin were also analyzed and quantified. All data represented as mean ± SEM (*n* = 6 per group, independent experiments in each figure).

**Figure 5 antioxidants-10-01086-f005:**
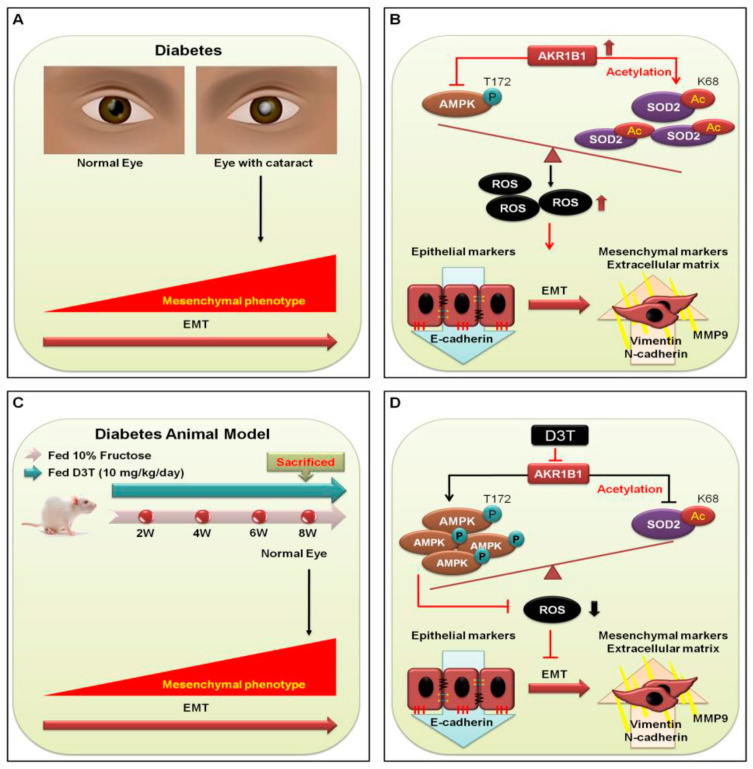
A summary of EMT function and D3T−downregulated AKR1B1 signaling pathway. (**A**) Diagram depicting the signal transduction pathways in DM (+) and DM (−) cataract patients. (**B**), In DM (−) cataract patients, p-AMPK^T172^ normal function is to regulate SOD2 activity to remove unwanted ROS. However, in DM (+) cataract patients, AKR1B1 inhibits the expression of p-AMPK^T172^ and promotes SOD2 acetylation to form AcSOD2 which leads to failure of ROS removal. The accumulated ROS induces oxidative stress, which in turn induces EMT progression, leading to the downregulation of epithelial marker and upregulation of mesenchymal markers (red line). In addition, N-cadherin and MMP9 expressions are significantly higher in DM (+) cataract patients. (**C**) WKY rats fed 10% fructose in drinking water for 8 weeks; or WKY rats fed 10% fructose in drinking water and administered D3T (10 mg/kg/day) via gavage for 8 weeks. (**D**) Fructose requires AKR1B1 to increase ROS generation (black line), and ROS accumulation further inhibits the expression of p-AMPK^T172^ and promotes SOD2 acetylation to form AcSOD2. In consequence, accumulated ROS stress induces EMT progression (red line). In contrary, D3T is capable of inhibiting the effect of fructose-activated AKR1B1 signaling cascade.

**Table 1 antioxidants-10-01086-t001:** Demographics and Baseline Clinical Characteristics of the Study Participants.

	Control Group (*n* = 6)	DM Group (*n* = 6)	*p* Value
Age (years)	66.0 ± 3.5	66.8 ± 6.1	0.871 ^a^
Sex (male: female)	2:4	2:4	1.000 ^b^
BMI	24.0 ± 2.7	25.3 ± 2.6	0.522 ^a^
HBA1C	-	9.0 ± 1.4	-

Haemoglobin A1c (HbA1c) levels were determined in patients with DM but without diabetic retinopathy (DM group). BMI represented body mass index. Values represented mean ± SEM; ^a^
*p* values were derived from Mann–Whitney U test; ^b^
*p* values were estimated by Fisher’s exact test.

**Table 2 antioxidants-10-01086-t002:** General characteristics of the 3 groups of rats.

Parameter/Group	Control	Fructose 8 W	Fructose 8 W + D3T
Systolic blood pressure (mm Hg)	100.6 ± 2.0	161.4 ± 4.2 **	114.1 ± 2.9 ^##^
Fasting serum glucose (mg/dL)	98.7 ± 6.1	150.0 ± 2.0 **	159.7 ± 6.8 **
dHDL (mg/dL)	67.8 ± 1.1	58.8 ± 2.2	74.8 ± 0.9 ^##^
Triglycerides (mg/dL)	77.8 ± 7.8	156.0 ± 8.7 **	148.7 ± 9.0 **

The serum triglyceride (TG), fasting serum fructose, systolic blood pressure (SBP), and high-density lipoprotein (HDL) cholesterol levels were determined in rats fed fructose or D3T for 8 weeks. The values are shown as the means ± SEM (n = 6 per group, separate experimental groups); ** *p* < 0.01 vs. the control group; ^##^
*p* < 0.01 vs. the fructose group.

## Data Availability

All data generated or analysed during this study are included in this published article.

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
