# Peer review of "3H-1,2-Dithiole-3-Thione Protects Lens Epithelial Cells against Fructose-Induced Epithelial-Mesenchymal Transition via Activation of AMPK to Eliminate AKR1B1-Induced Oxidative Stress in Diabetes Mellitus"

_antioxidants, 2021, doi:10.3390/antiox10071086_

Round 1
Reviewer 1 Report
Authors report on studies of 3H-1,2-dithiole-3-thione (D3T) in affecting epithelial-mesenchymal transition (EMT) induced by receptor of advanced glycation end products (AGE), in conjunction with augmented aldose reductase- (AKR1B1-) enhanced ROS generation. D3T administration prevented fructose-induced hypertension but did not improve fructose-mediated metabolic defects. Nevertheless, D3T reduced AKR1B1-induced EMT for rats with fructose-induced type 2 diabetes mellitus. Finally, authors revealed that D3T ameliorates AKR1B1-mediated EMT through the Snail/Slug signaling pathway.
The derived conclusions are supported bz the results, however in the revised manuscript the two main aspects must be improved: i) for non-expert readres all involved relationships should be stated in the Introduction and in small introductory explanation sentences within the Results. The whole picture is shown in the Figure 5, however, it must be clear what was known before and what is the contribution of the current manuscript.
Therefore, in the revised Introduction, authors need to explain more a working hypothesis, summarizing mechanistically relationships between the key players and clearly distinguish, what was already published and what aspects/details had to be solved in this manuscript. This should be done in a way not only suitable for the informed reader and specialists.
Minor:
The Discussion should not start like an advertisment for the drug. The consequences for public health should be of course discussed, however later in the text of Discussion.
Author Response
Comments and Suggestions for Authors
Authors report on studies of 3H-1,2-dithiole-3-thione (D3T) in affecting epithelial-mesenchymal transition (EMT) induced by receptor of advanced glycation end products (AGE), in conjunction with augmented aldose reductase- (AKR1B1-) enhanced ROS generation. D3T administration prevented fructose-induced hypertension but did not improve fructose-mediated metabolic defects. Nevertheless, D3T reduced AKR1B1-induced EMT for rats with fructose-induced type 2 diabetes mellitus. Finally, authors revealed that D3T ameliorates AKR1B1-mediated EMT through the Snail/Slug signaling pathway.
- The derived conclusions are supported bz the results, however in the revised manuscript the two main aspects must be improved:i) for non-expert readres all involved relationships should be stated in the Introduction and in small introductory explanation sentences within the Results. The whole picture is shown in the Figure 5, however, it must be clear what was known before and what is the contribution of the current manuscript.Therefore, in the revised Introduction, authors need to explain more a working hypothesis, summarizing mechanistically relationships between the key players and clearly distinguish, what was already published and what aspects/details had to be solved in this manuscript. This should be done in a way not only suitable for the informed reader and specialists.
Reply:
Thank you for your kind suggestion. We agree with the reviewer`s suggestion about the need to explain a working hypothesis, summarizing mechanistic relationships between the key players and clearly distinguishthe new findings of this study. Therefore, we have reorganized the Introduction section and added detailed informationas follows: (P1-3).
Introduction
Introduction
Diabetes mellitus (DM) can lead to pathologies in many tissues in the eye structure, bearing the nature of systemic chronic metabolic disease and microangiopathiccharac-teristic (1). Diabetes has been shown to be a risk factor for age-related (AR) cataract, it has been predicted that by the year 2030, >400 million people in the world will have diabetes (2). Aging population and longer life expectancy mean that DM will exceed 33% of the 65 years old and elder population by 2050, and global increase in AR cataract is also expected (3). In general, DM patients show tendency of up to five times more likely to develop cataract, particularly during early stage (4, 5). At present, other than removal of cataract, surgeries to treat cataract continue to increase every year (6). Although technological and medical advancement have greatly improved surgical outcome, the quality of vision in DM patients who undergo cataract removal remains unpredictable. Many studies showed complications from cataract surgery in DM patients, such as diabetic macular oedema (ME), postoperative ME, diabetic retinopathy progression, and posterior capsular opacification (2).
Evidence indicated that clinical antidiabetic drug treatment did not yield any clear association with lower cataract risks (7). Furthermore, studies found a statistically sig-nificant positive trend for association between diabetes duration and risk of cataract, and yielded increased relative risks for development of a cataract with higher HbA1c level (7). There is accumulating evidence that dietary sugar-mediated generation of advanced glycation end products (AGEs) and activation of the receptor for advanced glycation end products (RAGE), both of which are senescent protein derivatives that result from the auto-oxidation of glucose and fructose, contribute to metabolic syndrome (8, 9). More-over, impaired antioxidant ability of epithelial lenses caused increased susceptibility to oxidative stress in diabetic patients. Predominantly, the antioxidant enzyme, superoxide dismutase (SOD) in the lens degrades superoxide radicals (O2-) into H2O2 and oxygen. As a result, decreased activity of antioxidant enzymes, uncontrolled activation of AR, and accumulation of AGEs have been considered as signs hallmarks for cataract in diabetic patients (10, 11).
Previous study showed that the NADPH-dependent aldo-ketoreductase, also known as AKR1B1 in human, is a catalyst for converting glucose to sorbitol in the polyol pathway, was suggested to be critical for cataract development (12). Mechanistically, AKR1B1 is involved in retinopathy, cataractogenesis, neuropathy, and cardiovascular disease, which are part of diabetic complications (13). The inhibition of AKR1B1 has been an attractive approach for the treatment and management of diabetic complications. Evidence showed that AKR1B1 overexpression in the lens of transgenic mice led to anterior subcapsular cataracts development even in the absence of diabetes and hyperglycaemia (14). Additionally, it was demonstrated that AKR1B1 inhibitors were capable of suppressing lens epithelial cells` proliferation (14). Apart from AKR1B1 involvement, epithelial to mesenchymal cells transition (EMT) in lens` epithelial cells is also suggested to cause posterior capsule opacification (PCO), a common post-surgical complication resulted from cataract surgery (15).
The intermediary metabolism is regulates glucose and fatty acid uptake in cells, and it requires the serine/threonine protein kinase, AMP-activated protein kinase (AMPK). Under various stress conditions such as glucose starvation, hypoxia, ischemia, and oxi-dative damage, AMPK is activated to alter cellular energy level (16). In response to the stresses aforementioned, AMP:ATP ratio is changed through phosphorylation of Thr172 in the catalytic α subunit activation loop in the AMPK (17). Since AKR1B1 was shown to promote diabetic complications and play important role in oxidative stress during dia-betes, AKR1B1 inhibition was able to attenuate alcoholic liver disease by AMPK activation and oxidative stress modulation (18). Chang et al. reported that AR knockout increased the expressions levels of AMPK, SIRT1, and ATP in both normal and fatty livers after ma-jorhepatectomy and IR injury (19).
3H-1,2-Dithiole-3-thione (D3T) is a member of the 1,2-dithiole-3-thione-compounds naturally occurring in cruciferous vegetables. Among all 1,2-dithiole-3-thiones, D3T is the most potent in inducing tissue defences against oxidative and inflammatory stress(20). D3T potentiates AGE-induced ROS formation suppressed by inhibition of protein kinase C and NADPH oxidase (21). We previously reported that AKR1B1 overexpression abolishes AMPK activation, thereby increasing acetyl superoxide dismutase 2 (AcSOD2) and RAGE-induced EMT, and is an important factor involved in the cataract pathogenesis (5).
In this study, we aimed to investigate whether D3T reduces EMT in the fruc-tose-induced DM through an AKR1B1-enhanced ROS generation. We showed that fructose-induced DM, AKR1B1 overexpression would abolish AMPK activation, thereby increasing AcSOD2 and RAGE-induced EMT in the LECs, even without DCs. D3T downregulated AKR1B1-induced AMPK and caused an AcSOD2 imbalance, thereby decreasing EMT in the LECs, which is known to be a potentially important factor in the pathogenesis of cataracts in rats with fructose-induced DM. These novel findings suggest that the D3T may be a candidate for the pharmacological prevention of cataracts in patients with DM.
Minor:
- The Discussion should not start like an advertisment for the drug. The consequences for public health should be of course discussed, however later in the text of discussion.
Reply:
Thank you for your kind suggestion. We have reorganized the Discussion sectionas follows,and included theconsequences for public healthinthe revise manuscript: (P11-13).
Discussion
Diabetes is a major public health problem associated with the increased risk for cardiovascular disease, leading cause of death worldwide (31). Studies in yeast suggested that uncontrolled blood glucose elevation is the major risk factor for DM cataract development (32). Lens` opacity and oxidation (33), ROS formation, and sorbitol accu-mulation through AR conversion of glucose (34) have been linked to blood glucose ele-vation and lens protein glycosylation during cataract development. Since AGEs are found to play major role in lens degeneration, low level of AGE foods might be beneficial for counteracting cataract formation (35). This study confirms that AGEs and MMP9 levels are significantly upregulated in DM (+) cataract patients (Figure 1). We previously showed that RAGE induced NADPH oxidase expression in the fructose-induced type 2 DM lens, even though there was no cataract formation (36). Current evidence supports the view that cataract formation may involve EMT (37). In regards to EMT, Du et al. reported that oxidative stress increases AR activation and AGE formation, driving EMT in early stages of diabetic cataract formation (37). Furthermore, Korol at al. reported that MMPs are required for EMT in fibrosis of anterior subcapsular cataract (ASC) mouse models (24). Besides, AKR1B1 is an NADPH-dependent aldo-ketoreductase, well-known as a catalyst for glucose conversion to sorbitol in the polyol pathway, which has been suggested to be essential for cataract development (12). Furthermore, it was demonstrated that AKR1B1 inhibition led to downregulation of lens epithelial cells` proliferation and EMT (14). Interestingly, this study finds that AKR1B1 was capable of inducing EMT protein expression in the lens of rats with fructose-induced type 2 DM without developing cataract, and AKR1B1 treatment reversed the potentiation effect of D3T (Fig. 2, 4). These results suggest that NADPH blockage by D3T prevented AKR1B1-induced oxidants` generation, downregulated E-cadherin, and Snail/Slug signalling pathway.
D3T, a cyclic sulfur-containing dithiolethione derived from cruciferous vegetables, protects cells from oxidative lesions by production of antioxidants, and this antioxidative effect is mainly through activation of the nuclear factor (erythroid-derived 2)-like 2 (Nrf2) signalling pathway (38). As a transcription factor, Nrf2 regulates the expression of several target genes, including those encoding glutathione synthesizing enzymes, quinonereductases, glutathione re-ductase (GR), as well as NADPH generating enzymes such as glucose-6-phosphate de-hydrogenase (G6PD) (39). AGEs utilize RAGE to modulate gene expression and mediate superoxide generation upon NADPH oxidase activation (40). When Nrf2-mediated NADPH is downregulated, NAD(P)H oxidases (major source of AGE-induced oxidative stress) seize this opportunity to increase ROS such as superoxide. In order to prevent cell damage by ROS, Nrf2 activator, D3T, induces GSH synthesis (41). Global gene co-expression analysis in fasted animals (energy deficient animals) enabled the identification of AMPK as a potential Nrf2 kinase. AMPK plays a key role in sensing and modulating energy homeostasis by monitoring energy status (the AMP/ATP ratio) in the eukaryotic cells (42). Wang S et al., deletion of AMPK2 enhanced NADPH oxidase subunit expression and increased oxidative stress in vascular endothelial cells (43). Furthermore, Misra and Chakrabarti found that the severity of type II diabetes was effectively reduced via enhancing AMPK activity (27). Besides, Tsubota et al. showed that downregulated AMPK signalling pathway was involved in lens inflammation in the diabetic mice (44), indicating that AMPK activity is involved in accumulation of superoxide and other ROS. In addition, the inhibition of AKR1B1 attenuated alcoholic liver disease by activating AMPK and modulating oxidative stress (18). Our results demonstrated that 3-nitrotyrosine (3-NT) and AcSOD2 levels were elevated significantly in LECs, whereas p-AMPKT172 level was reduced significantly in the lenses obtained from rats with fructose-induced type 2 DM. Additionally, D3T treatment enhanced AMPK activity, reduced ROS productions, and decreased acetylation of SOD2 in the lenses obtained from the rats with fructose-induced type 2 DM (figure 3). Joo et al. also demonstrated that AMPKThr172 directly phosphorylates Nrf2S550 to facilitate its accumulation in the nucleus (46). Their findings identified AMPK as a novel kinase which activates Nrf2 through nuclear accumulation, leading to the conclusion that AMPK is an energy-sensing enzyme in cells with increased defense mechanism against antioxidants for cell survival (45). Collectively, these results suggest that strategies aimed at combating oxidative stress by downregulating the endogenous antioxidant defence system via AMPK may prevent cataract in patients with DM. Available scientific evidence suggests D3T as potential therapeutic agent for cataract associated with type 2 DM, since the D3T treatment can be administered through the eye drops.
According to the International Diabetes Federation, 415 million people had diabetes in 2015 and this number is expected to reach 642 million by 2040, with >55% of cases ex-pected to occur in Asia (46). High incidence of diabetes poses serious public health and financial burdens. The annual medical cost for diabetes treatment is estimated to increase from $802 billion to $1,452 billion in 2040 (47). Therefore, identifying new and easy-to-access preventive compound(s) that can alleviate health complications associated with diabetes is imperative. Evidence indicated that clinical antidiabetic drug treatment did not yield any clear associations with lower cataract risks. . In addition, our studies found that D3T reduces EMT in the fructose-induced DM through an AKR1B1-enhanced ROS generation. We also found that in fructose-induced DM, AKR1B1 overexpression abolished AMPK activation, leading to increased AcSOD2 and RAGE-induced EMT in the LECs in the absence of DCs. On the contrary, D3T downregulated AKR1B1-induced AMPK, causing an imbalance of AcSOD2, and reduced EMT in the LECs of fructose-induced DM rats. These novel findings suggest that D3T is a potential candidate for pharmacological prevention of cataracts in patients with DM.
In conclusion, AKR1B1 overexpression inhibited AMPK activation, thereby increasing AcSOD2 and RAGE-induced EMT in the LECs of diabetic cataracts. Regulation of AMPK activation is crucial for pathogenesis of diabetic cataracts (Figure 5B). AKR1B1 overexpression abolished AMPK activation, leading to increased AcSOD2 and RAGE-induced EMT in the LECs of rats with fructose-induced DM, even without diabetic cataracts. D3T downregulated AKR1B1-induced AMPK activation and AcSOD2 generation, leading to decreased EMT in the LECs (Figure 5D). These novel findings deepen our understanding of D3T as an important candidate for pharmacological intervention to treat cataracts in DM patients.
References:
- Skarbez K, Priestley Y, Hoepf M, Koevary SB. Comprehensive Review of the Effects of Diabetes on Ocular Health. Expert Rev Ophthalmol. 2010;5(4):557-77.
- Kiziltoprak H, Tekin K, Inanc M, Goker YS. Cataract in diabetes mellitus. World J Diabetes. 2019;10(3):140-53.
- Boyle JP, Thompson TJ, Gregg EW, Barker LE, Williamson DF. Projection of the year 2050 burden of diabetes in the US adult population: dynamic modeling of incidence, mortality, and prediabetes prevalence. Popul Health Metr. 2010;8:29.
- Saxena S, Mitchell P, Rochtchina E. Five-year incidence of cataract in older persons with diabetes and pre-diabetes. Ophthalmic Epidemiol. 2004;11(4):271-7.
- Wu TT, Chen YY, Chang HY, Kung YH, Tseng CJ, Cheng PW. AKR1B1-Induced Epithelial-Mesenchymal Transition Mediated by RAGE-Oxidative Stress in Diabetic Cataract Lens. Antioxidants (Basel). 2020;9(4).
- Drinkwater JJ, Davis WA, Davis TME. A systematic review of risk factors for cataract in type 2 diabetes. Diabetes Metab Res Rev. 2019;35(1):e3073.
- Becker C, Schneider C, Aballea S, Bailey C, Bourne R, Jick S, et al. Cataract in patients with diabetes mellitus-incidence rates in the UK and risk factors. Eye (Lond). 2018;32(6):1028-35.
- Guglielmotto M, Aragno M, Tamagno E, Vercellinatto I, Visentin S, Medana C, et al. AGEs/RAGE complex upregulates BACE1 via NF-kappaB pathway activation. Neurobiol Aging. 2012;33(1):196 e13-27.
- Miller A, Adeli K. Dietary fructose and the metabolic syndrome. Curr Opin Gastroenterol. 2008;24(2):204-9.
- Kandarakis SA, Piperi C, Topouzis F, Papavassiliou AG. Emerging role of advanced glycation-end products (AGEs) in the pathobiology of eye diseases. Prog Retin Eye Res. 2014;42:85-102.
- Sadi G, Eryilmaz N, Tutuncuoglu E, Cingir S, Guray T. Changes in expression profiles of antioxidant enzymes in diabetic rat kidneys. Diabetes Metab Res Rev. 2012;28(3):228-35.
- Snow A, Shieh B, Chang KC, Pal A, Lenhart P, Ammar D, et al. Aldose reductase expression as a risk factor for cataract. Chem Biol Interact. 2015;234:247-53.
- Grewal AS, Bhardwaj S, Pandita D, Lather V, Sekhon BS. Updates on Aldose Reductase Inhibitors for Management of Diabetic Complications and Non-diabetic Diseases. Mini Rev Med Chem. 2016;16(2):120-62.
- Zablocki GJ, Ruzycki PA, Overturf MA, Palla S, Reddy GB, Petrash JM. Aldose reductase-mediated induction of epithelium-to-mesenchymal transition (EMT) in lens. Chem Biol Interact. 2011;191(1-3):351-6.
- Wormstone IM, Wang L, Liu CS. Posterior capsule opacification. Exp Eye Res. 2009;88(2):257-69.
- Lin SC, Hardie DG. AMPK: Sensing Glucose as well as Cellular Energy Status. Cell Metab. 2018;27(2):299-313.
- Jia J, Bissa B, Brecht L, Allers L, Choi SW, Gu Y, et al. AMPK, a Regulator of Metabolism and Autophagy, Is Activated by Lysosomal Damage via a Novel Galectin-Directed Ubiquitin Signal Transduction System. Mol Cell. 2020;77(5):951-69 e9.
- Herzig S, Shaw RJ. AMPK: guardian of metabolism and mitochondrial homeostasis. Nat Rev Mol Cell Biol. 2018;19(2):121-35.
- Li CX, Wang HW, Jiang WJ, Li GC, Zhang YD, Luo CH, et al. The Inhibition of Aldose Reductase Accelerates Liver Regeneration through Regulating Energy Metabolism. Oxid Med Cell Longev. 2020;2020:3076131.
- Li YR, Jia Z, Zhu H. 3H-1,2-Dithiole-3-Thione as a Potentially Novel Therapeutic Compound for Sepsis Intervention. React Oxyg Species (Apex). 2019;8(22):202-12.
- Pazdro R, Burgess JR. The antioxidant 3H-1,2-dithiole-3-thione potentiates advanced glycation end-product-induced oxidative stress in SH-SY5Y cells. Exp Diabetes Res. 2012;2012:137607.
- Kim YK, Kim SG, Kim JY, Heo YK, Park JC, Oh JS. Histologic evaluation of a retrieved endosseous implant: a case report. Int J Periodontics Restorative Dent. 2013;33(1):e32-6.
- Tan AG, Kifley A, Holliday EG, Klein BEK, Iyengar SK, Lee KE, et al. Aldose Reductase Polymorphisms, Fasting Blood Glucose, and Age-Related Cortical Cataract. Invest Ophthalmol Vis Sci. 2018;59(11):4755-62.
- Stitt AW. Advanced glycation: an important pathological event in diabetic and age related ocular disease. Br J Ophthalmol. 2001;85(6):746-53.
- Ramana KV. ALDOSE REDUCTASE: New Insights for an Old Enzyme. Biomol Concepts. 2011;2(1-2):103-14.
- Hashim Z, Zarina S. Advanced glycation end products in diabetic and non-diabetic human subjects suffering from cataract. Age (Dordr). 2011;33(3):377-84.
- Chen YY, Wu TT, Ho CY, Yeh TC, Sun GC, Kung YH, et al. Dapagliflozin Prevents NOX- and SGLT2-Dependent Oxidative Stress in Lens Cells Exposed to Fructose-Induced Diabetes Mellitus. Int J Mol Sci. 2019;20(18).
- Du L, Hao M, Li C, Wu W, Wang W, Ma Z, et al. Quercetin inhibited epithelial mesenchymal transition in diabetic rats, high-glucose-cultured lens, and SRA01/04 cells through transforming growth factor-beta2/phosphoinositide 3-kinase/Akt pathway. Mol Cell Endocrinol. 2017;452:44-56.
- Korol A, Pino G, Dwivedi D, Robertson JV, Deschamps PA, West-Mays JA. Matrix metalloproteinase-9-null mice are resistant to TGF-beta-induced anterior subcapsular cataract formation. Am J Pathol. 2014;184(7):2001-12.
- Li KR, Yang SQ, Gong YQ, Yang H, Li XM, Zhao YX, et al. 3H-1,2-dithiole-3-thione protects retinal pigment epithelium cells against Ultra-violet radiation via activation of Akt-mTORC1-dependent Nrf2-HO-1 signaling. Sci Rep. 2016;6:25525.
- Kumar H, Kim IS, More SV, Kim BW, Choi DK. Natural product-derived pharmacological modulators of Nrf2/ARE pathway for chronic diseases. Nat Prod Rep. 2014;31(1):109-39.
- Wautier MP, Chappey O, Corda S, Stern DM, Schmidt AM, Wautier JL. Activation of NADPH oxidase by AGE links oxidant stress to altered gene expression via RAGE. Am J Physiol Endocrinol Metab. 2001;280(5):E685-94.
- Stochelski MA, Wilmanski T, Walters M, Burgess JR. D3T acts as a pro-oxidant in a cell culture model of diabetes-induced peripheral neuropathy. Redox Biol. 2019;21:101078.
- Hardie DG, Ross FA, Hawley SA. AMPK: a nutrient and energy sensor that maintains energy homeostasis. Nat Rev Mol Cell Biol. 2012;13(4):251-62.
- Wang S, Zhang M, Liang B, Xu J, Xie Z, Liu C, et al. AMPKalpha2 deletion causes aberrant expression and activation of NAD(P)H oxidase and consequent endothelial dysfunction in vivo: role of 26S proteasomes. Circ Res. 2010;106(6):1117-28.
- Misra P, Chakrabarti R. The role of AMP kinase in diabetes. Indian J Med Res. 2007;125(3):389-98.
- Kubota S, Ozawa Y, Kurihara T, Sasaki M, Yuki K, Miyake S, et al. Roles of AMP-activated protein kinase in diabetes-induced retinal inflammation. Invest Ophthalmol Vis Sci. 2011;52(12):9142-8.
- Joo MS, Kim WD, Lee KY, Kim JH, Koo JH, Kim SG. AMPK Facilitates Nuclear Accumulation of Nrf2 by Phosphorylating at Serine 550. Mol Cell Biol. 2016;36(14):1931-42.
- Li X, Xu Z, Ji L, Guo L, Liu J, Feng K, et al. Direct medical costs for patients with type 2 diabetes in 16 tertiary hospitals in urban China: A multicenter prospective cohort study. J Diabetes Investig. 2019;10(2):539-51.
- Zinman B. The International Diabetes Federation World Diabetes Congress 2015. Eur Endocrinol. 2015;11(2):66.

Reviewer 2 Report
The authors presented an interesting and well-written paper, however, several issues need to be addressed before publication.
Major:
Most of the protein quantification is based on MFI, to support the author claims western blots should be also performed.
Please note that the most recent reference date from 2019. Please update the reference list.
Minor:
Line 140. "For oral administration, animals were randomly assigned to five groups, each group contained 6 animals: 1) control: water; 2) fructose: 10% fructose water for 8 weeks; 3) D3T: 142 fructose + D3T (10 mg/kg/day) for 8 weeks." it should be 3 and not 5 groups.
How type 2 DM was induced in the rats, describe in detail the protocol used?
Figure 1. It is unclear how patient samples were collected. Please improve the quality of the pictures and add a scale bar. Replace the bar graphic with dot plots. If material available a western blot would be a more robust quantitative method.
Table 1. On the legend is mentioned that HbA1c was quantified in controls but the values are not present on the table.
Figure 2. Western blot quantification should be performed to substantiate the MFI values.
Line 272-290: Results should be described accordingly to the figure. In all the paragraph there is only one mention of the figure.
Line 313: The legend mentioned that immunoblots were performed but none is present in the figure.
Please add the origin of the D3T and mentioned also the solvent used
Author Response
Comments and Suggestions for Authors
The authors presented an interesting and well-written paper, however, several issues need to be addressed before publication.
Major:
- Most of the protein quantification is based on MFI, to support the author claims western blots should be also performed.
Reply:
Thank you for your valuable comment. We agree with the reviewer`s point that western blot would be a more robust quantitative method. However, in patients who had undergone phacoemulsification, the protein content obtained from the lens capsules was insufficient (less than 10 ng/µL) for western blot. Therefore, we performed immunofluorescent staining instead. However, we performed western blot analysis to double confirm the MFI values in the animal model as showed in figure 4.
- Please note that the most recent reference date from 2019. Please update the reference list.
Reply:
Thank you for your valuable suggestion.We have updated the reference list and also included references from 2020 (Ref no. 5, 17, 19, and 30) in the revise manuscript.
Minor:
- Line 140. "For oral administration, animals were randomly assigned to five groups, each group contained 6 animals: 1) control: water; 2) fructose: 10% fructose water for 8 weeks; 3) D3T: 142 fructose + D3T (10 mg/kg/day) for 8 weeks." it should be 3 and not 5 groups.
Reply:
We apologize for this oversight. In the revised manuscript, we have modified the sentence from “five groups” to “three groups” in the Material and Methods section. (P4, L11).
- How type 2 DM was induced in the rats, describe in detail the protocol used?
Reply:
In our previous study, we revealed a significant elevation of serum LDL and triglyceride concentrations in the animals of the fructose group compared with those of the control group. Moreover, we showed higher fasting blood glucose and fructose levels in the animals of the fructose group. Furthermore, the serum insulin and homeostatic model assessment as an index of insulin resistance (HOMA-IR) levels were elevated in the fructose-fed rats, whereas the dHDLlevel was significantly decreased, suggesting that fructose induced type 2 DM in these animals (1, 2).The experimentaldetailsare asfollows:
For oral administration,animals were randomly assigned to three groups, each group contained six animals: 1) control group: WKY rats fed normal drinking water; 2) fructose group: consisting of WKY rats fed 10% fructose in drinking water for 8 weeks; 3) The fructose+D3T group: composed of WKY rats fed 10% fructose in drinking water and administered D3T (10 mg/kg/day) via gavage for 8 weeks. We have added this detailed description in the Materials and Method section of the revised manuscript (P4, Line 12-15).
- Figure 1. It is unclear how patient samples were collected. Please improve the quality of the pictures and add a scale bar. Replace the bar graphic with dot plots.
Reply:
We apologize for the unclear description of patient sample collection in the previous draft. In brief, the central flap of anterior lens capsule consists of a single layer of lens epithelium with apices directed inward, and a basal laminar membrane that isolates the lens constituents. When patients underwent phacoemulsification, the central flap of anterior lens capsule was removed. As shown in the figure 1, we obtained the anterior region of lens capsule (central epithelium region, the red area) from patient and used for IF staining. We have improved the quality of the images and also add a scale bar in the figure 1 of revised manuscript(P6).
Figure 1.Expression of matrix metalloproteinase 9 (MMP9) and advanced glycation end products (AGEs) in the lens of DM (+) and DM (-) cataract patients. (A-B) Representative images of MMP9- positive (green), AGEs-positive (red) cells, and cell nuclei counterstained with DAPI (blue) in the lens of epithelial tissue from cataract DM (+) and DM (-) patients. Data represented mean ± SEMs (n = 6 per group, separate experimental groups in each figure). * P < 0.05.
- If material available a western blot would be a more robust quantitative method.
Reply:
Thank you for your valuable comment.We agree with the reviewer`s point that western blot would be a more robust quantitative method. However, in patients who had undergone phacoemulsification, the protein content obtained from the lens capsules was insufficient (less than 10 ng/µL) for western blot. Therefore, we performed immunofluorescent staining instead.
- Table 1. On the legend is mentioned that HbA1c was quantified in controls but the values are not present on the table.
Reply:
Thank you for highlighting this and we apologize for this oversight. We have modified the sentence from “Haemoglobin A1c (HbA1c) levels were determined in patients without diabetes mellitus (DM; control group) and in patients with DM but without diabetic retinopathy (DM group)” to “Haemoglobin A1c (HbA1c) levels were determined in patients with DM but without diabetic retinopathy (DM group).” in the table 1 legends. (P6, Line 7-8).
- Figure 2. Western blot quantification should be performed to substantiate the MFI values.
Reply:
Thank you for your insightful suggestion.We agree withyour point, and therefore, we performed western blot analysis to investigate if D3T ameliorated AKR1B1-mediated EMT via the Snail/Slug signalling pathway. As shown in figure 4, immunoblotting analysis of proteins extracted from the LECs demonstrated that D3T treatment inhibited the expression of Aldose reductase and AcSOD2 and increased AMPK levels in rats fed with fructose for 8 weeks (Fig. 4A). In fructose-fed animals, level of EMT-related transcription factors such as Snail and Slug were increased, whereas D3T administration reversed this effect (Fig. 4B). However, D3T treatment decreased E-cadherin expression in rats fed with fructose for 8 weeks (Fig. 4C). These results indicated that D3T reduced AKR1B1-induced EMT via activating AMPK in the LECs derived from rats with fructose-induced type 2 DM. (P9, Line 21-29).
- Line 272-290: Results should be described accordingly to the figure. In all the paragraph there is only one mention of the figure.
Reply:
Thank you for your comment and we apologize for the inadequate description in the previous draft. We have revised the sentence as follows: “As shown in figure 3, animals fed with fructose exhibited downregulated levels of p-AMPKT172and upregulated levels of 3-NT and AcSOD2; however, co-administration of D3T reversed these effects(Fig. 3A and B).(P9, Line 6-9)
- Line 313: The legend mentioned that immunoblots were performed but none is present in the figure.
Reply:
We are very sorry for this oversight and have removed “immunoblots” from the concerned sentence and havemodified the sentence appropriately.
- Please add the origin of the D3T and mentioned also the solvent used.
Reply:
Thank you for your comment. We have added the information regardingthe vendors from whom D3T and dimethyl sulfoxide (DMSO) were procured,in the Materials and Methods section of the revise manuscript. (P3, line 30-32)
References:
- Cheng PW, Lin YT, Ho WY, Lu PJ, Chen HH, Lai CC, et al. Fructose induced neurogenic hypertension mediated by overactivation of p38 MAPK to impair insulin signaling transduction caused central insulin resistance. Free Radic Biol Med. 2017;112:298-307.
- Yeh TC, Liu CP, Cheng WH, Chen BR, Lu PJ, Cheng PW, et al. Caffeine intake improves fructose-induced hypertension and insulin resistance by enhancing central insulin signaling. Hypertension. 2014;63(3):535-41.

Round 2
Reviewer 2 Report
The authors reply to most of my comments.
Please do replace the bar graphic with dot plots in Figure 1.
The authors only showed HBA1C in Diabetic patients, how the controls were selected and what was done to confirm the normal glycemia levels of these patients.
Author Response
- Please do replace the bar graphic with dot plots in Figure 1.
Apply:
Thanks for the comment of the reviewer. We have already replace the bar graphic with dot plots in Figure 1 of the revise manuscript.
- The authors only showed HBA1C in Diabetic patients, how the controls were selected and what was done to confirm the normal glycemia levels of these patients.
Apply:
Thanks for the comment of the reviewer. These samples underwent surgical operation at the Department of Ophthalmology, Kaohsiung Veterans General Hospital, Kaohsiung, Taiwan. The study will collect the lens capsule of DM (+) and DM (-) cataract patients. The age range of DM (+) and DM (-) cataract patients are 60≧ and ≦72 age. The patients were classified into 2 groups dependent on International Classification of Diseases-9 (ICP-9): patients without DM and hypertension (Group 1) and patients with DM but without DR and hypertension (Group 2). All of clinical sample obtained from clinical patients at the Kaohsiung Veterans General Hospital, Kaohsiung, Taiwan. Informed consent was obtained from all patients. All of the data included age, sex, BMI, HbA1C (only DM patients have) and clinical specimens were previously collected and were anonymized from Biobank before analysis. The sample sizes amount 0.1x0.1 cm2, discard sample less than 0.1 cm2. All clinical samples and information for the patients were obtained from biobank. Patients without DM and hypertension (Group 1) did not showed fasting glucose data and HbA1C in the clinical; therefore, we did not to confirm the normal glycemia levels of these patients.
